# Decoupling Intrinsic and Measurement Trends: A Crucial Consideration in Time Series Causal Discovery

## Abstract

In the realm of time series data, it is common to encounter time trends, which manifest as a function concerning time within a given data span. Time trends can be classified into intrinsic (real) and measurement (false) trends. Intrinsic trends are inherent to the underlying mechanisms of the variables, while measurement trends are essentially measurement errors unique to the observed values (e.g., an increase in diagnosed thyroid nodule patients due to enhanced medical techniques, despite a stable incidence rate over time). Measurement trends can critically influence the results of a variety of causal discovery methods and hence, necessitate elimination prior to causal analytic procedures. In this study, we introduce a novel framework capable of detecting all trend-influenced variables and distinguishing between intrinsic and measurement trends, called Trend Differentiator (TrendDiff). This approach consists of two primary steps: trend variable identification and trend type differentiation. The first step leverages Constraint-based Causal Discovery from heterogeneous/Nonstationary Data (CD-NOD) to identify variables with trends. Following this, we utilize the structure characteristics to differentiate between intrinsic and measurement trends. Experimental results on various synthetic scenarios and real-world data sets are employed to demonstrate the efficacy of our methods.

## 1 Introduction

Emerging in the early 1990s, causal discovery algorithms have undergone substantial growth in the past two decades (Spirtes & Zhang, 2016). These algorithms strive to infer causal relationships from purely observational data, serving as a valuable instrument in situations where randomized controlled trials are rendered impractical due to ethical concerns, financial constraints, and other obstacles. Standing at the intersection of explosive data volumes and advancements in computational capabilities, a surge in theoretical and applied causal research has ensued. Hitherto, causal discovery methods have been employed across various disciplines, such as climatology, healthcare, and economics, among others(Ebert-Uphoff & Deng, 2012; Shen et al., 2020; Hall-Hoffarth, 2022). Yet the rapid accumulation of data presents not only exhilarating possibilities but significant challenges in the domain of causal discovery.

A prevalent challenge is the presence of time trends, frequently encountered in time series data. As articulated by Phillips (2005), "No one understands trends, but everyone sees them in the data" (Phillips, 2005). While previous efforts have extensively examined the impact of time trends on the performance of conventional statistical algorithms (White & Granger, 2011; Wu et al., 2007), the effects on causal discovery methodologies remain unexplored. Given that the definition of time trends is still contentious, to be more precise, time trends are defined as a function concerning time within a given data span here. Based on the origin of these trends, they are classified into two categories: intrinsic (real) and measurement (false) trends. Intrinsic trends are inherent to the fundamental mechanisms governing the variables (e.g., global warming, the temperature is really increasing), whereas measurement trends are essentially observation errors unique to the recorded values (e.g., an observed increase in diagnosed thyroid nodule patients due to enhanced medical techniques, despite a stable real incidence rate over time (Davies & Hoang, 2021), see **Figure 1** ).

These two types of trends originate from distinct sources, exert disparate impacts, and necessitate differential treatment in the context of causal discovery.

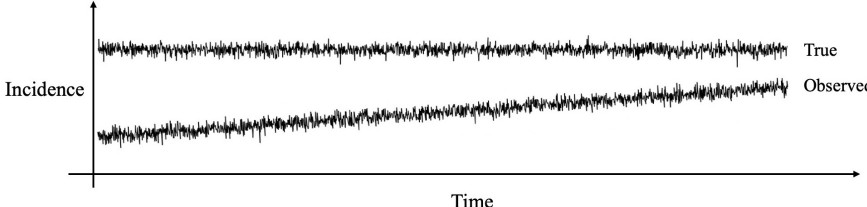

**Figure 1:** The true and observed incidence of the thyroid nodule along time – a typical example of measurement trends.

However, there is this impression – time trends, be it an intrinsic trend, or a measurement trend, should be removed before analyses – which is not accurate. Undoubtedly, measurement trends, being a form of measurement error, necessitate removal. Consider constraint-based causal discovery methods, which rely on conditional independence tests; measurement trends introduce two issues for constraint-based algorithms: 1. the dependence between measurement-trend variables and their neighbors weakens with increasing trends; 2. the conditional independence given the measurement-trend variables vanishes, yielding increasing dependence (Scheines & Ramsey, 2016; Zhang et al., 2017a). As illustrated in **Figure 2**, the measurement trend in $X_2$ not only affects its dependence with $X_1$ and $X_3$ (the dependence decreases with trends), but also causes $X_2$ to fail in separating $X_1$ and $X_3$. Analogous phenomena transpire for another measurement-trend variable $X_3$. The causal network identified by constraint-based methods diverges significantly from the ground truth in such scenarios. As noted in earlier research regarding measurement error in causal discovery, measurement trends not only influence constraint-based causal algorithms but also extend their impact to other methodologies, including those based on functional causal models (Zhang et al., 2017a). Conversely, intrinsic trends are integral components of the variables and mechanisms, facilitating the identification of underlying causal relationships. Removal of intrinsic trends would decrease the signal-to-noise ratio, leading to lower detection power, and thus should be avoided. Consequently, discerning between intrinsic and measurement trends and eliminating the latter is crucial before conducting causal discovery analyses.

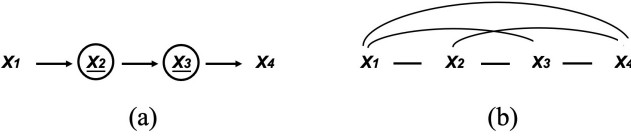

(a)                                          (b)

**Figure 2:** An illustration of how ignoring measurement trends in causal discovery may lead to spurious connections by constraint-based methods. (a) The true causal graph (including measurement-trend variables $\underline{X_2}$ and $\underline{X_3}$, the true values of which are not observable). (b) The estimated skeleton on the observed data. Note: the circled underlined variables $\underline{X_2}$ and $\underline{X_3}$ in (a) are real values, while $X_2$ and $X_3$ in (b) are observed values with measurement trends. )

In the present study, we assume the underlying causal structure to be a directed acyclic graph (DAG) containing variables exhibiting time trends, either intrinsic or measurement. Our objective is to devise a principled framework capable of identifying trend-influenced variables and distinguishing those with measurement trends from those exhibiting intrinsic trends. The paper is structured as follows: Section 2 defines the research question using DAG. Section 3 outlines our methodology for pinpointing variables exhibiting time trends, encompassing both intrinsic and measurement trends. In Section 4, we delve deeper into the techniques employed to distinguish between intrinsic and measurement trends. Together, these two sections offer a thorough exposition of the method Trend Differentiator (TrendDiff) used to identify and differentiate variables based on their time trends. Finally, an array of simulation studies under various scenarios and a real-world application are presented in section 5, substantiating the efficacy of our approach.

## 2 PARAMETERIZING TIME TRENDS

To put intrinsic and measurement trends clearer, we resort to structural equation models (SEMs), where each variable $V_i$ is formulated as a function of its direct causes and an error term $\varepsilon_i$. Here $\varepsilon_i$ encapsulates all other unmeasured causes of $V_i$ and $\varepsilon_i$ of variables are independent of each other. **Figure 3** shows the structures of intrinsic and measurement trends, respectively. Figure 3 (a) illustrates a straightforward model featuring a causal chain from $X_1$ to $X_2$ to $X_3$. Each variable is associated with a structural equation, and the model can be parameterized by assigning exact functions to $f(V_i)$, as well as a joint normal distribution to $\varepsilon_1, \varepsilon_2, \varepsilon_3 \sim \mathcal{N}(\mu, \Sigma^2)$. In this case, $\Sigma^2$ is diagonal, reflecting the independence among the error terms $\varepsilon_1$, $\varepsilon_2$, and $\varepsilon_3$. Regardless of the functions and free parameter values assigned, the model in Figure 3 (a) exhibits conditional independence: $X_1 \perp\!\!\!\perp X_3 \,|\, \mathbf{X}_2$.

In Figure 3 (b), we present the same model as in Figure 3 (a) but with an added intrinsic trend $T_2$ affecting $X_2$. The intrinsic trend $T_2$ impacts the generation of $X_2$ and is an inherent part of its underlying mechanisms. In this case, the observed and real values of $X_2$ are identical. The added intrinsic time trend is able to go into the causal network through $X_2$ without altering the original causal structure. Consequently, a trend in $X_3$ can be observed, which arises due to the influence of $T_2$. In Figure 3 (c), we depict the same model but with true values $\underline{X_2}$ being "measured" as $X_2$, accompanied by a measurement trend $T_2$. In this case, the real and observed values of $X_2$ differ. The measurement trend $T_2$ is present only in the observed $X_2$. Due to the collider at $X_2$, $T_2$ cannot influence the real values $\underline{X_2}$ and is unable to propagate through the original causal network. As previously mentioned, here the measurement trend $T_2$ essentially represents a form of measurement error, which can adversely affect the performance of causal discovery algorithms.

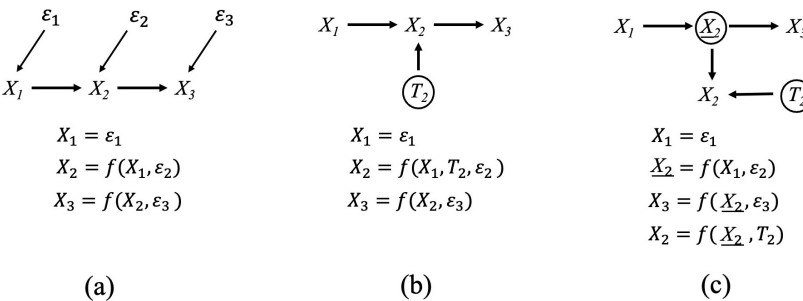

**Figure 3:** An illustration of causal models for variables with intrinsic and measurement trends and corresponding equations. (a) A three-variable chain graph without trend). (b) $X_2$ with an intrinsic trend. (c) $X_2$ with a measurement trend, $\underline{X_2}$ represents the true values of $X_2$. Note: variables without a circle are observed variables, while those with a circle are hidden variables. To make the graph clearer, we omitted $\varepsilon_1, \varepsilon_2, \varepsilon_3$ in Figure 3 (b) and (c)

## 3 PHASE 1: DETECTION OF TIME-TREND VARIABLES AND CAUSAL STRUCTURE RECOVERY

### 3.1 ASSUMPTIONS

Adopting a more relaxed version of causal sufficiency, this work assumes pseudo causal sufficiency. In causal discovery, the causal sufficiency assumption posits that all common causes (confounders) of the observed variables are included in the data set. The presence of time trends in data, however, may violate this assumption. Time trends typically emerge from intricate, compounded factors. As a statistical expedient, these factors are collectively considered, predicated on the combined effect being expressible as a mathematically smooth function of time when quantitatively represented. Time trends across distinct variables can be interrelated due to specific types of unobserved confounders. Consequently, we merely assume that these confounders, if any, are fixed at each time point within data exhibiting time trends, which is referred to as pseudo causal sufficiency.

Assuming that the observed data are independently identically distributed (I.I.D), this work concentrates on instantaneous or contemporaneous causal relationships, and the strength of the causal relations does not change over time. As a consequence, time-delayed causal relations, specifically autoregressive models, are not explicitly explored. Nevertheless, it is worth noting that our framework can be naturally generalized to encompass time-delayed causal relations in time series, akin to how constraint-based causal discovery has been adapted to manage time series data (see, e.g. (Chu et al., 2008)).

Let $\{g_l(C)\}_{l=1}^{L}$ represent the set of confounders (potentially empty). Additionally, we posit that for each $V_i$, the local causal process can be depicted by the SEM:

$$V_i = f_i\left(\mathrm{PA}^i, \mathbf{g}^i(C), \theta_i(C), \varepsilon_i\right) \tag{1}$$

Here, $\mathbf{g}^i(C) \subseteq \{g_l(C)\}_{l=1}^{L}$ signifies the set of confounders influencing $V_i$ (an empty set when no confounder is present behind $V_i$ and any other variable), while $\theta_i(C)$ represents the effective parameters within the model, also presumed to be functions of $C$. Moreover, $\varepsilon_i$ denotes a disturbance term, independent of $C$ and exhibiting non-zero variance (i.e., the model is non-deterministic). The mutual independence of $\varepsilon_i$ is also assumed.

In this work, we consider $C$ as a random variable, yielding a joint distribution over $\mathbf{V} \cup \{g_l(C)\}_{l=1}^{L} \cup \{\theta_m(C)\}_{m=1}^{n}$. We assume that this distribution adheres to the Markov and faithfulness properties with respect to the graph resulting from the subsequent modifications to $G$ (which, as a reminder, represents the causal structure over $\mathbf{V}$): add $\{g_l(C)\}_{l=1}^{L} \cup \{\theta_m(C)\}_{m=1}^{n}$ to $G$, and for each $i$, add an arrow from each variable in $\mathbf{g}^i(C)$ to $V_i$ and add an arrow from $\theta_i(C)$ to $V_i$. This extended graph is denoted as $G^{aug}$. Evidently, $G$ is merely the induced subgraph of $G^{aug}$ over $\mathbf{V}$. Importantly, leaf nodes — those devoid of descendants — manifest characteristics indistinguishable when they are with either an intrinsic or a measurement trend. Hence, we assume that trend variables are not positioned as leaf nodes.

## 3.2 DETECTION OF TIME-TREND VARIABLES AND CAUSAL STRUCTURE RECOVERY

In this section, we use the Constraint-based Causal Discovery from heterogeneous/Nonstationary Data (CD-NOD) to detect variables exhibiting time trends and subsequently deduce the causal network for $\mathbf{V} \cup \{C\}$. The core concept hinges on using the (observed) variable $C$ as a surrogate for the unobserved $\{g_l(C)\}_{l=1}^{L} \cup \{\theta_m(C)\}_{m=1}^{n}$. In essence, we utilize $C$ to encapsulate the $C$-specific information. Under the assumptions detailed in Section 3.1, it becomes feasible to deploy conditional independence tests on the combined set of $\mathbf{V} \cup \{C\}$ to detect variables with time trends and recover the structure. This is achieved by Algorithm 1 and supported by Theorem 1.

In Algorithm 1, we first construct a complete undirected graph, denoted $U_C$, which incorporates both $C$ and $\mathbf{V}$. In Step 2 of the algorithm, the decision regarding whether a variable $V_i$ exhibits a time trend is contingent upon the conditional independence between $V_i$ and $C$, given a subset of other variables. If a time trend is present in $V_i$, then the module of $V_i$ evolves in conjunction with $C$. Consequently, the probability distribution $P\left(V_i \mid \mathrm{PA}^i\right)$ will not remain constant across different values of $C$. As a result, $V_i$ and $C$ are conditionally dependent regardless of any subset of other variables. Based on this rationale, we assume that if $V_i \perp\!\!\!\perp C \mid \mathrm{PA}^i$, then there should be no time trend in $V_i$. Conversely, if this assumption does not hold, then we claim to detect variables with time trends. After this step, all variables linked to $C$, referred to as "$C$-specific variables", are considered to be with time trends. It's important to highlight that this step is characterized by high recall; however, its precision might exhibit slight variations. This precision is contingent on the number of no-trend variables possessing changing modules within the data set. Specifically, Algorithm 1 has been designed to effectively identify all variables exhibiting changing modules. While time-trend variables inherently exhibit a changing module, the reverse is not necessarily true. As a result, our categorization of "$C$-specific variables" also encompasses variables that, although devoid of trends, display changing modules. Given that our focus is refined to "$C$-specific variables" throughout Phase 2, this characteristic ensures Phase 1 is conservative. "$C$-specific variables" will usually equal to or larger than the true trend-variable set, thereby guaranteeing the comprehensive inclusion of every trend variable.

---

**Algorithm 1** Detection of Time-trend Variables and Recovery of Causal Structure

---

1: Build a complete undirected graph $U_{\mathcal{G}}$ on the variable set $\mathbf{V} \cup C$.
2: (Detection of time-trend variables) For each $i$, test for the marginal and conditional independence between $V_i$ and $C$. If they are independent given a subset of $\{V_k \mid k \neq i\}$, remove the edge between $V_i$ and $C$ in $U_{\mathcal{G}}$.
3: (Recovery of causal skeleton) For every $i \neq j$, test for the marginal and conditional independence between $V_i$ and $V_j$. If they are independent given a subset of $\{V_k \mid k \neq i, k \neq j\} \cup \{C\}$, remove the edge between $V_i$ and $V_j$ in $U_{\mathcal{G}}$.
4: (Orientation) For the obtained skeleton, orient it by standard orientation rules and distribution shift. After the orientation process, we can get the causal network for $\mathbf{V} \cup C$, called $G^{\text{phase1}}$.

---

Step 3 aims to discover the skeleton of the causal structure over V. It leverages the results from Step 2: if neither $V_i$ nor $V_j$ is adjacent to $C$, then $C$ does not need to be involved in the conditioning set. In practice, one may apply any constraint-based search procedures on $\mathbf{V} \cup C$, e.g., SGS and PC (Spirtes et al., 1993). Its (asymptotic) correctness is justified by the following theorem:

**Theorem 1:** *Given Assumptions made in Section 3.1, for every $V_i, V_j \in \mathbf{V}, V_i$ and $V_j$ are not adjacent in G if and only if they are independent conditional on some subset of $\{V_k \mid k \neq i, k \neq j\} \cup \{C\}$.*

Given that this segment is identical to the Constraint-based Causal Discovery from Heterogeneous/Nonstationary Data (CD-NOD), we refrain from delving into further details here. For a comprehensive explanation of the step 4 orientation procedure and the complete proof of Theorem 1, please refer to (Zhang et al., 2017b; Huang et al., 2020).

## 4 PHASE 2: UTILIZING STRUCTURAL DIFFERENCES TO DISTINGUISH BETWEEN INTRINSIC AND MEASUREMENT-TREND VARIABLES

In Phase 1, we procured the set of variables exhibiting time trends (those associated with $C$) as well as the causal network $G^{phase1}$ for $\mathbf{V} \cup C$. By constraining our analysis to only the "$C$-specific variables" while pinpointing intrinsic-trend variables, Phase 2 of our algorithm benefits from increased efficiency and a reduced risk of false positives. Besides, although the derived causal structure $G^{phase1}$ in Phase 1 might not be entirely accurate due to the existence of measurement trends, it serves as a foundational aid in differentiating types of trends. In phase 2, we demonstrate that by examining the different structures within causal networks, it is feasible to differentiate variables with intrinsic trends from those influenced by measurement trends.

### 4.1 DISTINGUISH BETWEEN INTRINSIC AND MEASUREMENT TRENDS BY $G^{phase1}$

As depicted earlier, intrinsic-trend variables do not change the causal network, whereas those variables characterized by measurement trends can induce structural alterations in causal discovery. Next, we delve into how a measurement-trend variable influences the causal structure of $G^{\text{phase1}}$ and leverage this understanding to partly distinguish between the two trend types.

**Figure 4** illustrates how a measurement-trend variable alters the output causal structure of Phase 1. In Figure 4(a), we depict a chain with a measurement trend in $X_2$. During Phase 1, the time index $C$ is integrated into our analysis to pinpoint all trend variables. Due to the presence of a measurement trend in $X_2$, a connection from $C$ to $X_2$ is established. Furthermore, based on the conditional independence observed in the actual structure Figure 4(a), we have $T \perp\!\!\!\perp X_3$ and, crucially, $T \not\perp\!\!\!\perp X_3|X_2$. By extension, because $C$ is a proxy for $T$, the relationships $C \perp\!\!\!\perp X_3$ and $C \not\perp\!\!\!\perp X_3|X_2$ should hold. The dependency dynamics between $X_1$ and $C$ follow suit. As a result, the Phase 1 structural outcome should be the one shown in Figure 4(b). It's worth noting that since the measurement trend $T$ is independent across all variables within the causal network, no arrow can stem from the measurement-trend variable to other variables in $G^{\text{phase1}}$. In essence, any linkage from a "$C$-specific variable" to other entities indicates an intrinsic trend.

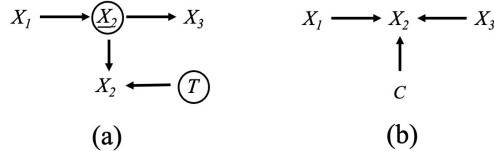

**Figure 4:** An illustration of how a measurement-trend variable alters the output causal structure of Phase 1. (a) the real structure with a measurement trend in $X_2$. (b) the output structure of (a) by the algorithm in Phase 1. $X_2$ represents observed values, $\underline{X_2}$ represents the true values of $X_2$. Note: variables without a circle are observed variables, while those with a circle are hidden variables.

In summary, we first employ the structure of $G^{\text{phase1}}$ to discern intrinsic-trend variables. A "C-specific variable" is deemed to exhibit an intrinsic trend if it possesses an arrow pointing to other variables in $G^{\text{phase1}}$.

## 4.2 DISTINGUISH BETWEEN INTRINSIC AND MEASUREMENT TRENDS BY FURTHER CONDITIONAL INDEPENDENCE TESTS

Having identified certain intrinsic-trend variables based solely on the structure of $G^{\text{phase1}}$, it becomes necessary to undertake additional conditional independence tests for further recognition of more intrinsic-trend variables. As illustrated in **Figure 3**, the children of time-trend variables serve as critical pivot points in their differentiation process. For variables with intrinsic trends (see Figure 3b), there is $T_2 \not\perp\!\!\!\perp X_3$ and $T_2 \perp\!\!\!\perp X_3|X_2$. Conversely, for variables with measurement trends (see Figure 3c), there is $T_2 \perp\!\!\!\perp X_3$ and $T_2 \not\perp\!\!\!\perp X_3|X_2$. Thus, the criterion for identifying an intrinsic-trend variable $X_2$ can be $T_2 \not\perp\!\!\!\perp X_3$ and $T_2 \perp\!\!\!\perp X_3|X_2$. Here $T_2$ is the trend of $X_2$ and $X_3$ is a child of $X_2$. Since the trend $T_2$ is not directly observable in this context. As an alternative, we employ the time index $C$ again, working as a suitable proxy for the unobservable trend. Therefore, the criterion is: $C \not\perp\!\!\!\perp X_3$ and $C \perp\!\!\!\perp X_3|X_2$.

The first row of **Figure 5** illustrates four scenarios of child variables that may arise when screening for the intrinsic-trend variable $X_1$. In Figure 5 (a), no trend is evident in the child variable $X_2$, allowing us to easily identify $X_1$ as an intrinsic-trend variable using our criterion. However, in Figure 4 (b)(c), the child variable $X_2$ exhibits intrinsic and measurement trends, respectively. Since trends are functions of time, time serves as a confounder (common cause) of trends $T_1$ and $T_2$. In these cases, the path from $T_1$ to $X_2$ via the confounder "time" cannot be blocked, as neither "time" nor $T_2$ is observable (we can obtain a surrogate for $T_2$, but it is insufficiently accurate to block the path). Consequently, we cannot distinguish variables with intrinsic trends from those with measurement trends when all child variables have trends. However, if the trend in the child variable $X_2$ originates from its other observable parent $X_3$, as depicted in Figure 4 (d), the intrinsic-trend variable $X_1$ is identifiable since we can block the path through "time" by conditioning on $X_3$.

For structures (b) and (c), first-order descendants (children) do not facilitate distinguishing trend types. However, can second-order descendants provide clarity? Will it help if structures similar to (a) or (d) emerge subsequent to (b) and (c)? The subsequent row illustrates potential second-order descendant structures for both (b) and (c). Although (b-1) and (b-2) remain non-identifiable, (c-1) and (c-2) can be discerned. The principles behind (c-1) and (c-2) align with those of (a) and (d), namely $C \not\perp\!\!\!\perp X_3$ and $C \perp\!\!\!\perp X_3|X_1$. It's noteworthy that structures (c-1) and (c-2) essentially represent (a) and (d) but with an added measurement-trend variable subsequent to the intrinsic-trend variable $X_1$ under examination. Extending this rationale, we can infer that all structures obtained by adding $n$ measurement-trend variables between $X_1$ and $X_2$ in structures (a) and (d) can theoretically be identified, where $n$=0,1,2...

In summary, intrinsic-trend variables are discernible only when (1) the intrinsic-trend variable $X$ to be tested possesses at least one descendant variable $Y$ without trends (like structure (a)) or with trends stemming from other observable variables (like structure (d)); and (2) there are no other intrinsic-trend variables on the path from $X$ to $Y$. Nevertheless, the performance deteriorates in reality as the number of measurement-trend variables between $X_1$ and $X_2$ increases, due to the amplification of noise with increasing distance. To maintain accuracy, this study restricts its focus

to first-order scenarios, wherein $X_2$ is a direct descendant, namely a child of $X_1$. Algorithm 2 for Phase 2 is provided in Appendix A.1. Combining Algorithm 1 and Algorithm 2, we can obtain the proposed Trend Differentiator (TrendDiff Algorithm).

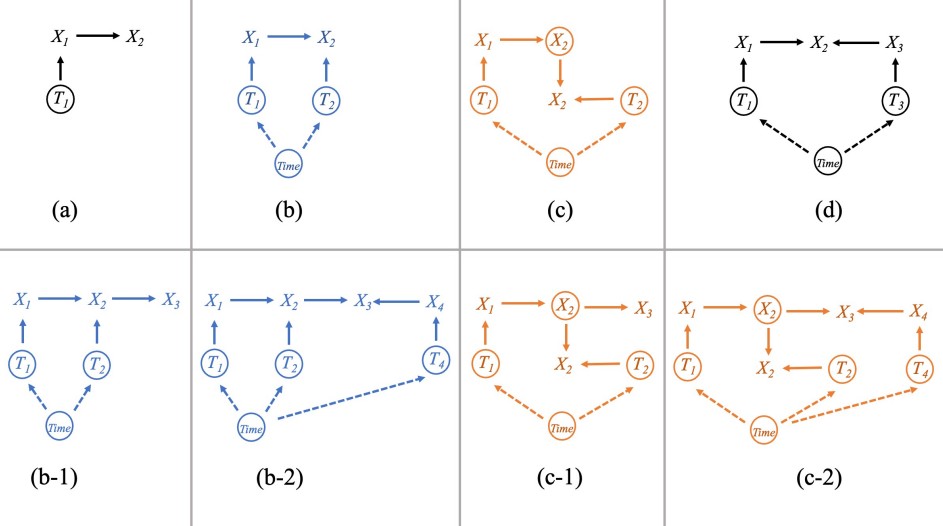

**Figure 5:** Different scenarios for descendants of intrinsic-trend variables. First raw: Four possible cases of intrinsic-trend variable's child nodes in causal networks. (a) Child node without trend. (b) Child node with an intrinsic trend. (c) Child node with a measurement trend. (d) Child node with a trend from other observable nodes. Second raw (b-1), (b-2), (c-1), and (c-2): Four possible cases of intrinsic-trend variable's second-order descendant for structure (b) and (c).

## 5 EXPERIMENTS

The proposed TrendDiff algorithm has been employed on a variety of synthetic and real-world data sets. We assessed the accuracy with which this method can pinpoint variables exhibiting intrinsic trends across diverse scenarios. Besides, we further contrasted the efficacy of causal discovery methodologies pre- and post-removal of measurement trends discerned by our techniques, thereby demonstrating the advantages of eliminating such trends.

### 5.1 SIMULATIONS

Algorithm performance is first evaluated by simulation data. We generated synthetic data according to the SEMs specified in Figure 8. More specifically, $V_1$, $V_5$, and $V_7$ have intrinsic trends, $V_2$, and $V_6$ have measurement trends. Time trends are defined as a sinusoid function of time, with periods $w$ randomly selected from the range(5,25). All relationships are nonlinear. We tried different noise types (Gaussian, Exponential, Gumbel), as well as different sample sizes (T = 600, 900, 1200, 1500). In each setting, we ran 50 trials. We tested the generated data with the proposed TrendDiff method and compared the results of PC algorithm before and after the removal of identified measurement trends.

**Figure 6** displays the simulation results. Figure 6 (a) presents the F1 score, precision, and recall of identified intrinsic-trend variables under varying data length T and noise type. The robustness of the proposed algorithm is evidenced by its consistent performance in Gaussian, Exponential, and Gumbel noise models. As the data length increases, there is a corresponding enhancement in performance. When data length equals to or above 1500, the algorithm demonstrates commendable efficiency, with the F1 score, precision, and recall all close to 0.9. The recall is a little bit lower compared with the precision. This discrepancy arises from our conservative approach, prioritizing the minimization of false-positive intrinsic-trend variables, as they present greater detrimental consequences than false negatives. Figure 6 (b) contrasts the efficacy of the PC algorithm in reconstructing the original causal network using data pre and post-elimination of detected measurement

trends. Removal of these measurement trends notably bolsters the performance of the PC algorithm, with a pronounced enhancement in the F1 score and precision. Besides these tests, we also generated data from random structures. We further used the data from random structure to evaluate the sensitivity of our approach towards data length, noise type, dimensionality (denoted by the number of nodes), and sparsity (defined by the degree considering edges in both directions). Since time trends may be approximately linear in some situations, we tested TrendDiff performance for linear-trend scenarios as well. Our method displayed stability across varying conditions, the results of which can be found in the Appendix. These results further establish its robustness and adaptability.

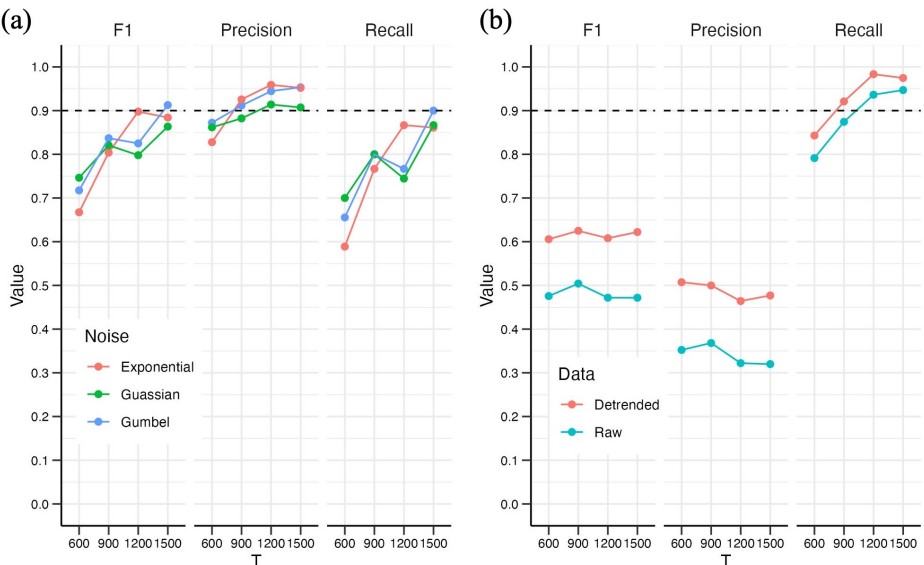

**Figure 6:** Simulation performance. (a)Performance of identifying intrinsic-trend variables under varying conditions, measured in terms of F1 score, precision, and recall (higher values indicate better performance). (b) Performance of PC algorithm using data pre and post-elimination of detected measurement trends.

## 5.2 REAL DATA

We also applied the proposed approach to a real environmental health dataset. This dataset contains daily values of variables regarding air pollution, weather, and sepsis emergency hospital admission in Hong Kong for the period from 2007 to 2018. It is a typical dataset used to assess the interactions between environmental factors and human health. There are pronounced time trends in this data (**Figure 7a**), rendering it a good application example for TrendDiff algorithm. In our initial analysis, we applied TrendDiff to determine the intrinsic trend variables within the data. The outcome from Phase 1 (as detailed in Algorithm 1) indicates that sepsis emergency hospital admissions, $CO$, $O_3$, and $SO_2$ are variables exhibiting a trend, be it measurement or intrinsic. Subsequently, in our follow-up phase (Algorithm 2), we differentiated between measurement-trend and intrinsic-trend variables. It was discerned that $CO$, $O_3$, and $SO_2$ have intrinsic trends while the daily count of sepsis emergency hospital admissions stood out as the sole variable characterized by a measurement trend. This result is consistent with existing evidence. There have been heated discussions in top medical journals about the observed rise in sepsis cases. A prevailing consensus among researchers is that this uptick in sepsis incidences can be largely attributed to the refined definitions and enhanced coding practices for sepsis, rather than the real incidence increase (Rhee et al., 2017; Fleischmann-Struzek et al., 2018). As for the trio of variables recognized with an intrinsic trend — $CO$, $O_3$, and $SO_2$ — ample research has been conducted on their trends. However, none have ascribed these trends to measurement inaccuracies, supporting our results here (Wei et al., 2022). Beyond merely distinguishing two types of trends, we also conducted a comparison of causal discovery results, before and after eliminating the identified measurement trend. Here the time series causal discovery method Peter-Clark-momentary-conditional-independence plus (PCMCI+) was adopted (Runge, 2020). Utilizing this environmental health dataset from Hong Kong, our primary objective

was to delineate the environmental determinants linked to sepsis. As illustrated in **Figure 7(b)**, there are significant variations in outcomes contingent on the removal of the measurement trend. Initial analyses using raw data classified $CO$ and $SO_2$ as mitigating factors against sepsis. However, upon exclusion of the sepsis measurement trend, only temperature was pinpointed as a synchronous risk factor for sepsis. Though this analysis did not deal with factors like seasonality, the observed discrepancies highlight the paramount importance of detecting and addressing measurement trends in causal discovery analysis.

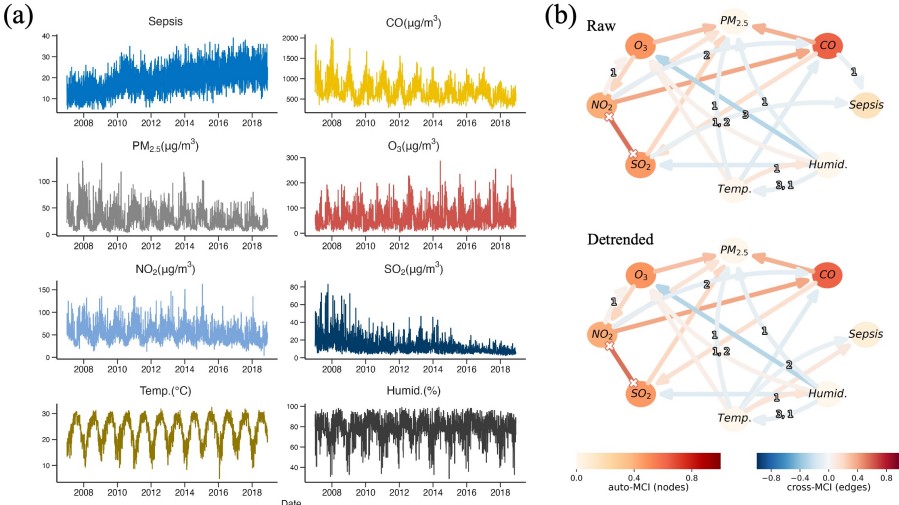

**Figure 7:** Evaluation of performance using a real-world dataset. (a) Depiction of time series variables. (b) Raw: discovery of structure from raw data by Peter-Clark-momentary-conditional-independence plus (PCMCI+). Detrended: discovery of structure after removal of identified measurement trends by PCMCI+.

## 6 CONCLUSION AND DISCUSSIONS

There has long been a pressing need for techniques to discern intrinsic trends from measurement trends. The proposed algorithm TrendDiff stands out as the first dedicated solution to this problem. Beyond its applicability in data pre-processing for causal discovery—as demonstrated in both simulated and real-world scenarios—its advantages are manifold. Firstly, by addressing measurement trends, which essentially is a kind of measurement error, data quality is enhanced. The adage "Garbage In, Garbage Out" underscores the pivotal role of data quality in application studies, a principle that spans multiple disciplines. This uplift in quality augments not only causal discovery but also the efficacy of a myriad of other methodologies. Secondly, the practical significance of this method is profound. For both entrepreneurs and investors, discerning genuine market trends from ephemeral ones is pivotal. Investing resources or capital in fake trends can culminate in substantial disappointment, given the absence of a genuine market fit. Algorithms tailored for distinguishing trend types play a crucial role in mitigating such risks.

In future work, we aim to solve the following questions: 1. How to further improve the performance of the current algorithm, especially when data length is limited? 2. What if a variable bears both intrinsic and measurement trends? Can we develop a method to distinguish the two types of trends within the same variable? 3. How to better remove the identified measurement trends? For linear trends, removal is straightforward. Yet, addressing nonlinear trends is more challenging, primarily because their exact form or shape is often unknown.

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

# A APPENDIX

## A.1 ALGORITHM 2

In phase 2, structural differences will be used to distinguish intrinsic-trend variables from measurement-trend ones. For each "$C$-specific variable" $X_i$ identified in phase 1 (these are variables with time trends), we first check if there is an outgoing link from this variable to others in the $G^{\text{phase1}}$. If there is, the variable is added to the intrinsic trend set. After this procedure, we further check each of the left "$C$-specific variable" $X_i$. We use time index $C$ as the surrogate of trend. Then conditional independence tests will be conducted between $C$ and each of $X_i$'s neighbor $X_j$ in the identified causal network $G^{\text{phase1}}$ under different conditional sets. Since the conditional sets are used to block paths via other parents of $X_j$ (as is shown in Figure 5 (d)), it will iterate on all combinations of $X_j$'s neighbors, rather than iterate on the combinations of all rest variables. For each pair of $X_i$ and $X_j$, once $C$ and $X_j$ are dependent conditions on any set $S_0$ and become independent conditions on the set $S_0 + X_i$, $X_i$ is identified as intrinsic-trend variables. After getting the intrinsic-trend set, we derive the set of measurement-trend variables by subtracting the intrinsic-trend variables from the "$C$-specific variables".

---

**Algorithm 2** Identify intrinsic-trend variables by structural differences

---

**Require:** Dataset $\mathbf{V}$, "$C$-specific variables" identified in phase 1, causal structure $G^{\text{phase1}}$ identified in phase 1, significance threshold $\alpha$, CI test $\text{CI}(X, Y, \mathbf{Z})$ returning $p$-value.
1: IntrinsicSet = $\emptyset$
2: **for all** $X_i \in$ "$C$-specific variables" **do**
3:     $\boldsymbol{\beta}$ = Causal Graph Matrix($G^{\text{phase1}}$)
4:     links = $\boldsymbol{\beta}_{i\cdot}$    **if** $1 \in$ links **then**           $\triangleright$ Check if $X_i$ has an outgoing link
6:     Store $X_i$ in IntrinsicSet
7:
8: RestSet = "$C$-specific variables" - IntrinsicSet
9: **for all** $X_i \in$ RestSet **do**
10:     TestNeighbors = Neighbors($X_i$) - IntrinsicSet
11:     **for all** $X_j \in$ TestNeighbors **do**
12:         JNeighbors = Neighbors($X_j$) - $X_i$
13:         **for all** $n \in$ Range(len(JNeighbors)) **do**
14:             **for all** $S_0 \in$ Combinations(JNeighbors, $n$) **do**
15:                 $S_1 = S_0 + X_i$
16:                 $C$ = Time index
17:                 $p_0 = \text{CI}(C, X_j, \mathbf{S_0})$
18:                 $p_1 = \text{CI}(C, X_j, \mathbf{S_1})$
19:                 **if** $(p_0 < \alpha)$ & $(p_1 > \alpha)$ **then**
20:                     Store $X_i$ in IntrinsicSet
21: MeasurementSet = "$C$-specific variables" - IntrinsicSet
22: **return** IntrinsicSet, MeasurementSet

---

**Note:** the Causal Graph Matrix in line 3 outputs a Causal Graph object, where $\boldsymbol{\beta}_{j,i} = 1$ and $\boldsymbol{\beta}_{i,j} = -1$ indicate i $\rightarrow$ j; $\boldsymbol{\beta}_{i,j} = \boldsymbol{\beta}_{j,i} = -1$ indicates i — j; $\boldsymbol{\beta}_{i,j} = \boldsymbol{\beta}_{j,i} = 1$ indicates i $\leftrightarrow$ j.

## A.2 FIXED STRUCTURE SIMULATION

The simulated data from the fixed structure is obtained following three steps: 1) Get the original fixed structure without trends. The structure used in this study to test our algorithm is shown in **Figure 8**. Figure 8 (a) is the SEMs and Figure 8 (b) is the visualization of the structure. 2) Embed intrinsic and measurement trends into the structure. To include both the identifiable structures (a) and (d) in Figure 5 in our simulation, we add intrinsic trends in $X_1$, $X_2$, and $X_7$. Besides, to mimic the properties of real-world data, we also add two measurement trends in $X_3$ and $X_6$. All trends are defined as smoothed functions of time, generating from $trend = \sin\left(\frac{w \cdot t}{T}\right)$. Here the period $w$ is randomly selected from a uniform distribution $\text{Unif}([5, 25])$, $T$ is data length, $t$ is time index. 3) Generate simulation data. Finally, we generate the simulation data based on the structure

constructed after steps (1) and (2). All relationships are nonlinear, with randomly selected 50% links using $f^{(1)}(x) = \left(1 - 4e^{-x^2/2}\right)x$ and 50% links using $f^{(2)}(x) = \left(1 - 4x^3 e^{-x^2/2}\right)x$. We tested our algorithm under the joint distribution of $\varepsilon_0, \varepsilon_1, ... \varepsilon_9 \sim$ (Gaussian, Exponential, Gumbel), respectively. We tried different sample sizes (T = 600, 900, 1200, 1500). We tested the proposed TrendDiff by the generated data in these different scenarios. In each setting, we ran 50 trials.

(a)

$X_0 = \varepsilon_0$          $X_5 = f(X_3, \varepsilon_5)$

$X_1 = \varepsilon_1$          $X_6 = f(X_5, \varepsilon_6)$

$X_2 = f(X_0, \varepsilon_2)$          $X_7 = f(X_5, \varepsilon_7)$

$X_3 = f(X_0, \varepsilon_3)$          $X_8 = f(X_6, \varepsilon_8)$

$X_4 = f(X_1, X_2, \varepsilon_4)$          $X_9 = f(X_7, \varepsilon_9)$

(b)

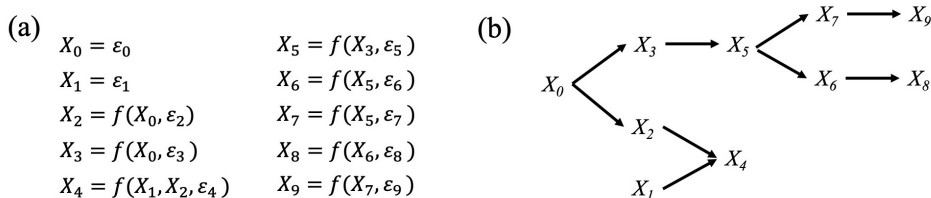

**Figure 8:** Data structure for fixed-structure simulation. (a) The SEMs according to which we added intrinsic and measurement trends and generated the simulated data. (b) The visualization of the structure. All relationships are nonlinear.

We also analyzed the outcomes of the PC algorithm both prior to and following the elimination of the measurement trends pinpointed by our TrendDiff. This comparison underscores the efficacy of our method in enhancing causal discovery performance by data pre-processing. In addressing variables with measurement trends, we employ the Savitzky-Golay filter to eliminate these trends. We subtract the Savitzky-Golay filtered trend from the raw data to obtain the detrended data. The Savitzky-Golay filter is a polynomial smoothing filter that essentially fits a polynomial of a given degree to a window of consecutive data points using a method of linear least squares. Once the polynomial is fitted to the data, the smoothed value or the derivative of the function can be obtained. The filter is commonly applied to noisy data in various fields, including analytical chemistry and signal processing. The two main parameters of it are the window size and the polynomial order. The window size dictates how many data points will be used for each polynomial fit and thus controls the overall smoothing, while the polynomial order controls the model complexity. Note that, if the trend variable is a leaf node, it is impossible to ascertain whether it has an intrinsic or a measurement trend. Therefore, we only address scenarios when the trend variables are not leaf nodes. Importantly, our strategy for finding variables with measurement trends is as follows: (1) Identify variables that exhibit trends, regardless of whether they stem from intrinsic factors or measurement errors (Phase 1), (2) Identify variables that possess intrinsic trends, and (3) The variables with measurement trends are then derived by subtracting the intrinsic-trend variables from the overall trend variables identified in Step (1). In this study, all variables labeled as "$C$-specific variables" exhibit trends. However, in real-world data, this set might also include nonstationary variables (those with changing modules) that don't necessarily have trends. Consequently, the set of measurement-trend variables, derived by subtracting the intrinsic-trend variables from the "$C$-specific variables", could have false positives. This implies that while the output for intrinsic-trend variables is highly accurate, the accuracy for the measurement-trend variables might be slightly compromised.

In our algorithm, employing a general, nonparametric conditional independence test is of paramount importance. The reason being, that the nature in which trends fluctuate over time remains unknown and is often profoundly nonlinear. In the present study, we leverage the kernel-based conditional independence test (KCI-test) (Spirtes et al., 1993) to capture these dependencies. Within the KCI test framework, the kernel width parameter, instrumental in constructing the kernel matrices, plays a pivotal role in determining performance outcomes. Our method's efficacy was assessed across various time lengths $T$ and kernel widths $w$ to choose optimal $w$. **Figure 9** elucidates the variations in performance based on distinct values of $w$ for evolving $T$. It is evident from the data that as $T$ ascends, there's an increase in performance. Furthermore, a kernel width of $w = 0.5$ consistently delivers commendable results irrespective of the $T$ value. These findings echo the guidelines proposed in the original KCI paper, which recommends: set $w$ to 0.8 for sample sizes $n \leq 200$, to 0.3 if $n > 1200$, and to 0.5 in all other instances. In alignment with this advice, our study adopted these kernel width settings.

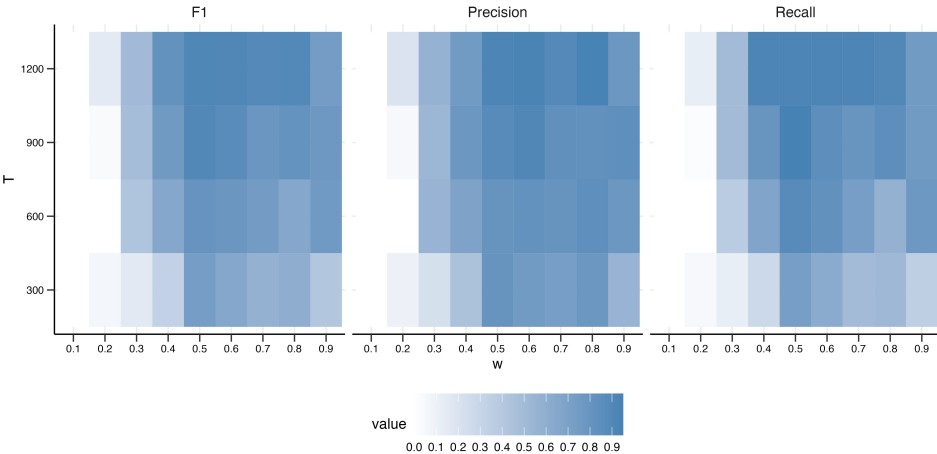

**Figure 9:** Parameter choosing results. Performance of our algorithm under different kernel width $w$ with changing data length $T$. The performance is evaluated using F1 score, precision, and recall.

### A.3 RANDOM STRUCTURE SIMULATION

We also tested our algorithm using simulated data based on random structures. There are three steps to this process: 1) We generated random graph G from the Erdös-Rényi (ER) random graph model, with edges added independently with equal probability. The degree, that is, the total number of edges linked with each node (in + out), is $d$. Given G, the weights of edges are drawn from $\mathrm{Unif}([-0.6, -0.2] \cup [0.2, 0.6])$ to obtain a weight matrix $W_0$. 2) Given $W_0$, intrinsic and measurement trends are randomly assigned to variables, with $W_0$ updated to $W$. Note that, only intrinsic trend structures like (a) and d in Figure 5 will be generated in this process, which means: a) no trend in leaf nodes; b) variables with trends are not adjacent. 3) Then we sampled $X = W^T X + z \in \mathbb{R}^d$ from noise model. Finally, we generated random datasets $\mathbf{X} \in \mathbb{R}^{n \times d}$ by generating T rows I.I.D. We considered different model setups for noise types, data length T, and the degree of sparsity to comprehensively test our algorithm. Performance is evaluated by F1 score, precision, and recall. Precision is the number of true positives divided by the number of true positives plus the number of false positives $P = \frac{TP}{TP+FP}$. Recall is the number of true positives divided by the number of true positives plus the number of false negatives $R = \frac{TP}{TP+FN}$. The F1 score is the harmonic mean of precision and recall. It provides a balance between precision and recall and is a good metric to consider if there is uneven class distribution $F1 = 2 \times \frac{P \times R}{P+R}$. For each scenario, all metrics are computed across all graphs from 50 realizations of the random graph-generating model at data length T in (600, 900, 1200, 1500).

**Figure 10** showcases the performance metrics—F1 score, precision, and recall—for identifying intrinsic-trend variables across different data lengths $T$ and noise types. Notably, the method proves robust across noise variations and, consistent with fixed structure results, performance improves with increasing data length. **Figure 11** provides further insights into our method's stability, demonstrating its resilience across a range of data dimensions and degrees of sparsity, where dimension is denoted by the number of nodes and sparsity is defined as the degree considering edges in both directions. When comparing the algorithm's performance in fixed-structure scenarios to those in random-structure situations, we observe a slight decrease in precision for the latter. However, the overall evaluation metric, the F1 score, remains consistently stable. **Figure 12** shows TrendDiff performance on data generated from random structures with linear trends. We measured the identification of intrinsic-trend variables across different data lengths T and noise types. TrendDiff excels in scenarios with linear trends.

### A.4 APPLICATION IN REAL-WORLD DATA

Besides simulation studies, we applied our algorithm to a real-world data set about environmental health as well. The data set contains daily values of variables regarding air pollution, weather, and

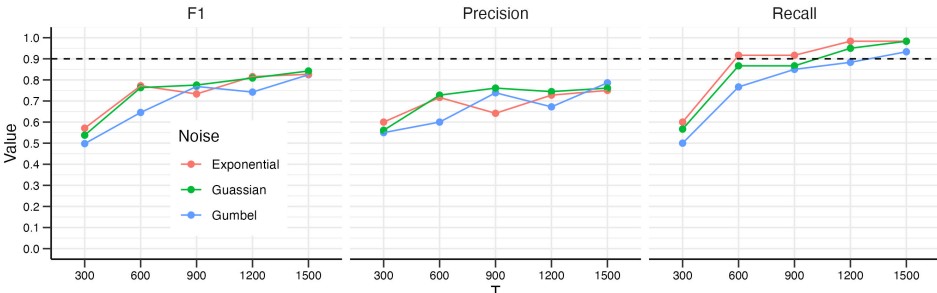

**Figure 10:** Performance evaluation on data generated from random structures with varying $T$ and noise type. We measure the identification of intrinsic-trend variables across different data lengths $T$ and noise types using F1 score, precision, and recall. Higher values denote better performance.

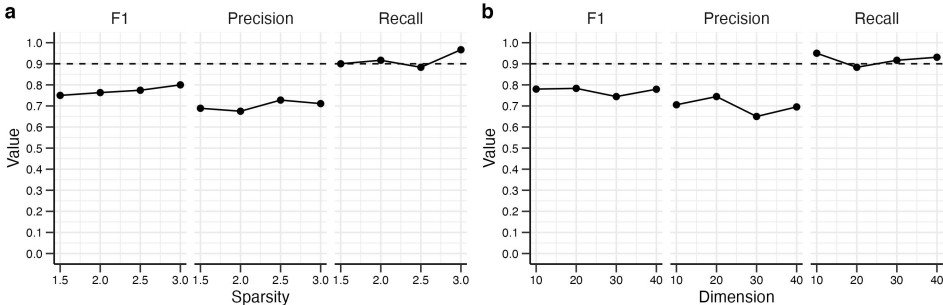

**Figure 11:** Performance evaluation on data generated from random structures with varying sparsity and dimension. (a) Performance under different sparsity levels. (b) Performance across varying dimensions.

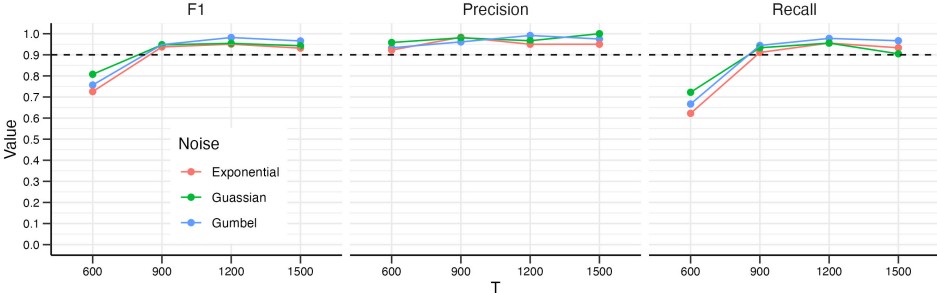

**Figure 12:** Performance evaluation on data generated from random structures with linear trends. We measure the identification of intrinsic-trend variables across different data lengths $T$ and noise types using F1 score, precision, and recall. Higher values denote better performance.

sepsis emergency hospital admission in Hong Kong for the period from 2007 to 2018. This data set is good for exploring the relationships between environmental factors and sepsis. Sepsis, alternatively referred to as septicemia or blood poisoning, is a life-threatening medical emergency when the dysregulated host response to infection injures its own tissues and organs (Singer et al., 2016). It is one of the leading causes of death and contributes significantly to preventable mortality (Organization et al., 2020). In 2017, 11.0 million sepsis-related deaths were reported globally, constituting 20% of all the annual deaths (Rudd et al., 2020). Understanding the relationships between environmental factors and sepsis risk provides a deeper insight into the underlying mechanisms through which environmental factors may predispose, trigger, or exacerbate sepsis conditions. This knowledge is not only pivotal for timely intervention but also offers a foundation for formulating targeted prevention strategies.

In our real-data application, we initially utilized the proposed method to identify sets of all trend variables, intrinsic-trend variables, and measurement-trend variables. We then compared these findings with existing literature on trend and measurement errors in environmental variables and sepsis data, reinforcing the accuracy of our algorithm. Furthermore, we employed the causal discovery algorithm, "Peter-Clark-momentary-conditional-independence plus (PCMCI+)", both before and after eliminating the detected measurement trends from the dataset. This comparative analysis of causal discovery performance serves to underscore the advantages of our algorithm as an effective data pre-processing tool.

Below we detail the PCMCI+ algorithm:

PCMCI+ belongs to the so-called constraint-based causal discovery methods family, which is based on conditional independence test(Runge, 2020). Here "PC" refers to the developers Peter and Clark, "MCI" means that the momentary conditional independence (MCI) test idea is added to the traditional PC algorithm, and "+" reminds users that it extends the earlier version of PCMCI to include the discovery of contemporaneous links(Runge et al., 2019). Like other causal graphic models, PCMCI+ works under the general assumptions of the causal Markov condition (each variable in the system is independent of its non-descendants, given its parent variables) and faithfulness (probabilistic information in data emerges not by chance but from causal structures) (Runge, 2018). On top of the general assumptions, two specific assumptions are also requested: causal stationarity (i.e., the causal links hold for all the studied time points) and causal sufficiency (i.e. measured variables include all of the common causes).

PCMCI+ algorithm starts with a skeleton discovery phase, which serves to remove the adjacencies due to indirect paths (mediation) and common causes (confounders). This phase can be divided into lagged stage and contemporaneous stage. The former is to identify lagged potential parents, and the latter is to identify contemporaneous potential parents and optimize identified lagged parents. In the lagged stage, for each variable $X_t^j$, a superset of lagged ($\tau > 0$) parents $\widehat{\beta_t^-}\left(X_t^j\right)$ is estimated with the iterative PC1 algorithm. In the contemporaneous stage, we iterate through subsets $\mathcal{S} \subset \boldsymbol{X_t}$ of contemporaneous adjacencies and remove adjacencies for all (lagged and contemporaneous) ordered pairs $\left(X_{t-\tau}^i, X_t^j\right)$ with $X_t^j \in \boldsymbol{X_t}$ and $X_{t-\tau}^i \in \mathbf{X}_t \cup \widehat{\beta_t^-}\left(X_t^j\right)$ if the MCI conditional independence holds: $\left(X_{t-\tau}^i \perp X_t^j \mid \mathcal{S}, \widehat{\beta_t^-}\left(X_t^j\right), \widehat{\beta_{t-\tau}^-}\left(X_{t-\tau}^i\right)\right)$. This skeleton discovery phase returns a skeleton of causal network of undirected relationships among the nodes.

Next in the orientation phase the contemporaneous links (lagged links can automatically be directed by time order) in the recognized skeleton will be oriented by the collider orientation stage and followed by the rule orientation stage. In collider orientation process, unshielded triples $X_{t-\tau}^i \to X_t^k \circ - \circ X_t^j$ (for $\tau > 0$) or $X_t^i \circ - \circ X_t^k \circ - \circ X_t^j$ (for $\tau = 0$) where $X_{t-\tau}^i, X_t^j$ are not adjacent would be oriented as collider structures if $X_t^k$ is not in the sepset $\left(X_{t-\tau}^i, X_t^j\right)$ according to the rule "none". Here sepset $\left(X_{t-\tau}^i, X_t^j\right)$ means the controlled variables when obtaining conditional independence of $X_{t-\tau}^i, X_t^j$. Besides the rule "none", another two rules "conservative" and "majority" can also be chosen in this stage. After that, three rules R1, R2, and R3 are followed to orient left links. R1 rule states that all unambiguous $X_{t-\tau}^i \to X_t^k \circ - \circ X_t^j$ can be oriented as $X_{t-\tau}^i \to X_t^k \to X_t^j$ since there is no collider left in this stage; in R2 rule, all $X_t^i \to X_t^k \to X_t^j$ structures with $X_t^i \circ - \circ X_t^j$ are

oriented as $X_t^i \to X_t^j$ to avoid circles. Finally, in R3 rule, for all unambiguous $X_t^i \circ - \circ X_t^k \to X_t^j$ and $X_t^i \circ - \circ X_t^l \to X_t^j$ where $X_t^k, X_t^l$ are independent and $X_t^i \circ - \circ X_t^j$, we orient $X_t^i, X_t^j$ as $X_t^i \to X_t^j$ to satisfy both the no-collider and no-circle rules. After the orientation process, we leave unoriented correlations as $\circ - \circ$ and conflicting correlations as $\times - \times$.

The main free parameters of PCMCI+ (in addition to the free parameters of the conditional independence tests) are the maximum time delay $\tau_{\max}$ and the significance threshold $\alpha_{\mathrm{PC}}$. We used 3 and 0.05 for these two parameters, respectively. In the output causal network produced by PCMCI+, a curved arrow represents a lagged causal relationship, with the lag day shown on the curve. A straight arrow means a contemporaneous association. A conflicting, contemporaneous adjacency "x-x" indicates that the directionality is undecided due to conflicting orientation rules. The link color refers to the cross-MCI value, which indicates the strength of the relationships. The node color denotes the auto-MCI value, representing how strong the autocorrelation is.

