# OpenReview forum: "Decoupling Intrinsic and Measurement Trends: A Crucial Consideration in Time Series Causal Discovery"
_ICLR.cc/2024/Conference — Submitted to ICLR 2024_

### Official Review · Reviewer_FN22 · 2023-10-30

**Soundness:** 3 good
**Presentation:** 3 good
**Contribution:** 2 fair
**Rating:** 5
**Confidence:** 3

**Summary:**

In this submission, the author(s) study the fundamental problem of causal discovery for time series data. The author(s) distinguish between into intrinsic time trends, which are inherent to the underlying mechanisms of the variables, and measurement trends, which are errors of the observations induced by the measurement itself. In this study, the author(s) introduce a novel framework capable of distinguishing between intrinsic and measurement trends. The author(s) provide theoretical analysis as well as experiments to showcase the performance of their algorithm.

**Strengths:**

In my opinion this paper is well-written. I believe that the distinction between inherent and measurement trends is relevant. I also believe that the experimental section is well-done.

**Weaknesses:**

In my opinion, the proposed algorithm is of a limited practical interest. The reason is that the proposed algorithm is based on conditional independence tests, which is generally quite difficult to evaluate, particularly if the conditioning random variable is high-dimensional. Furthermore, if I understand correctly, the author(s) assume faithfulness in their analysis. Faithfulness is an assumption that is common in the literature. However, this assumption may be unrealistic and it cannot be verified in practice.

**Questions:**

- Can you clarify how faithfulness is use in your analysis?
- Can you generalize your results beyond faithfulness?

---

### Official Review · Reviewer_Qn3u · 2023-10-31

**Soundness:** 2 fair
**Presentation:** 2 fair
**Contribution:** 2 fair
**Rating:** 3
**Confidence:** 4

**Summary:**

The authors propose an algorithm that can identify the variables influenced by time trends and further distinguish the intrinsic trends from the measurement trends, given time series observational data.

**Strengths:**

1. The objective pursued by the paper is engaging and is out of the motivation from the practices.
2. The authors conduct experiments on simulated and one real-world dataset.

**Weaknesses:**

1. Due to the absence of "the related work" section, it is challenging to know the most recent developments in the relevant field. Hence, it is hard to measure the significant contribution of the work.
2. Following the previous comment, there are no related baselines in the experiment section, making it challenging to evaluate the performance of the proposed method without comparison.
3. The paper relies heavily on the CD-NOD algorithm, and hence, it is better to explain how CD-NOD works briefly in the paper. Personally, the whole of section 3 seems to be redundant as it is similar to the existing content in the original papers about the CD-NOD algorithm. It will be more straightforward if the authors can introduce the CD-NOD algorithm and, on top of that, introduce their modifications. Otherwise, it is difficult to distinguish the novel contributions from the existing algorithm.
4. It is better to have an "Assumption" section to list all the assumptions needed instead of scattering assumptions throughout the paper.
5. The proposed algorithm works within a limited scope. Furthermore, the constraints are imposed directly on the DAG. Given observational data, the true DAG is unknown. Hence, the constraints can not be validated. That is, people will only know whether the algorithm works if they know the true DAG, which is exactly the hard problem people want to solve.
6. What is the computational complexity? Could you help me locate the related content if I missed it?
6. The experiments focus only on "first-order scenarios", further limiting the scope of the study.
7. The experimental results demonstrate the algorithm's capacity to identify 3 intrinsic-trend variables from 9 variables based on fixed structure simulations. However, the demonstration lacks comprehensive validation. Given weakness 5,  having fixed structure simulations seems to be a natural consequence of imposing constraints on DAG directly. Even so, could there be a class of structure that can be used in the simulation instead of a fixed one?

**Questions:**

1. $C$ is not defined when it first appears in the paper. What is $C$ so that $T$ is unobserved but $C$ is observed?
2. What do you mean by "$C$ is a proxy for $T$". Do you mean $T=g^i(C)$ in equation 1? If so, it is better to state it in the paper explicitly.
3. Assuming that the observed data are independently identically distributed (I.I.D) in a time series dataset may seem overly idealistic. Can you provide a rationale to support this assumption?
4. In the paper, "To maintain accuracy, this study restricts its focus to first-order scenarios, wherein X2 is a direct descendant...". What do you mean by "To maintain accuracy"? Does it mean the algorithm does not work well in other cases?
5. In the main paper, "V1, V5, and V7" are variables with intrinsic trends, however, in the appendix, "X1, X2, and X7" are with intrinsic trends. Is this a type?
6. Could you briefly explain how you eliminate the identified measurement trend after identifying the variables with measurement trends in the real-world dataset?
7. While the algorithm is primarily designed to identify variables with trends, it can, in fact, uncover the underlying causal graph during phase 1. As the true underlying causal graph is known in the simulations,  could you also present the results of the causal discovery, including F1, precision, and recall, by comparing the estimated causal graph with the true causal graph?

---

### Official Review · Reviewer_hzGy · 2023-11-07

**Soundness:** 2 fair
**Presentation:** 2 fair
**Contribution:** 2 fair
**Rating:** 3
**Confidence:** 4

**Summary:**

The paper proposes a causal discovery method to decouple intrinsic trends and measurement trends. The method consists of two steps. In the first step, it identifies variable with trends using CD-NOD. In the second step, it differentiate between the two types of trends.

**Strengths:**

Decoupling intrinsic trends and measurement trends could be important in practice.

Simulations are used to demonstrate the performance of the proposed method.

**Weaknesses:**

The proposed method is not very well justified. The main writing has been focused on providing intuition using examples rather than rigorously showing the soundness of the method.

The authors stated "Given that the definition of time trends is still contentious, to be more precise, time trends are defined as a function concerning time within a given data span here", which is however pretty vague to me. It would be helpful if it is defined using the SEM language.

The method assumes "these confounders, if any, are fixed at each time point within data exhibiting time trends". Can the authors provide a mathematical or graphical definition?

Variable "C" should be defined at the first time it appears in the paper. I believe it's the time index.

A discussion of the assumption "trend variables are not positioned as leaf nodes" will be helpful.

In "While time-trend variables inherently exhibit a changing module, the reverse is not necessarily true". I think the "reverse is not necessarily true" part needs to be explained.

For graphs like Figure 3(c), the main issue seems to come mostly from the measurement error. The intrinsic trends isn't that crucial. Can one simply ignore the intrinsic trends and treat this as a causal model with measurement errors, for which there is existing work? Would that allow us to identify the measurement trend variable?

**Questions:**

Is there any theoretical guarantee of the proposed method that can be made?

Is IID a reasonable assumption for time-series?

In "Consequently, the probability distribution P(V_i|PA^i) will not remain constant across different values of C", does that probability distribution have C integrated out? Or is it actually conditional on C, which is omitted in the notation?

---

### Meta-Review · Area_Chair_PzeP · 2023-12-06

**Metareview:**

The reviewers have several issues around the clarity, justification and soundness of the method. The authors chose not to upload a rebuttal to address these concerns.

**Justification For Why Not Higher Score:**

The authors chose not to upload a rebuttal to address reviewer concerns.

**Justification For Why Not Lower Score:**

N/A

---

### Decision · Program_Chairs · 2024-01-16

Reject